# The relationship between the source of oral health information and dental caries: Findings from Child Dental Health Survey 2013 in England

Ahmad Sofi-Mahmudi[1,2]*

1 National Pain Centre, Department of Anesthesia, McMaster University, Hamilton, ON, Canada,
2 Department of Health Research Methods, Evidence and Impact, McMaster University, Hamilton, ON, Canada

* sofima@mcmaster.ca, a.sofimahmudi@gmail.com

**Data Availability Statement:** All metadata and codes are publicly available from the OSF

## Abstract

### Objective

To determine the magnitude and shape of the relationship between dental caries experience and the source of oral health information in England.

### Methods

This was a cross-sectional study using the Child Dental Health Survey 2013 in England. Using a negative binomial model, the relationship between the number of decayed, missing, filled teeth (DMFT) of 12- and 15-year-old students and their primary source of oral health information was assessed. The sources of oral health information included parents, television, newspapers, the Internet, and social media. The adjusted model included age, sex, and the Index of Multiple Deprivation (IMD). R was used for data handling, analysis and reporting.

### Results

Overall, 2,372 children were assessed (48.7% female, 48.6% 12-year-old). For the majority, the primary source of oral health information was their parents (89.5%) followed by the Internet (43.4%). Over nine-tenth of the participants had a DMFT = 0. The adjusted model showed that the prevalence rate of DMFT for the children whose primary source of information is their parents (0.45) or television (0.62) is lower than 1. The prevalence rate for the Internet (1.17) and social media (1.67) was higher than 1, but they were removed from the final model due to being non-statistically significant. Age and deprivation had a direct relationship with the prevalence rate of DMFT, meaning that 15-year-olds and children from more deprived areas had a higher prevalence rate of DMFT.

repository (https://doi.org/10.17605/OSF.IO/2VAZM).

**Funding:** The author(s) received no specific funding for this work.

**Competing interests:** The authors have declared that no competing interests exist.

## Conclusion

Children whose primary source of oral health information was their parents or television had a lower DMFT. On the contrary, using the Internet or social media as the source of oral health information was associated with higher caries experience among schoolchildren.

## Introduction

Despite being largely preventable, dental caries persists as a substantial global health concern [1]. The 2022 Global Oral Health Status Report by the World Health Organization highlights that more than two billion individuals worldwide currently experience untreated dental caries in their permanent teeth [2]. In terms of economic impact, dental conditions resulted in both direct and indirect expenditures amounting to $544.41 billion in 2015, positioning it as the sixth most costly health condition on a global scale [3].

Effective communication and education about oral health may be crucial in preventing dental caries, particularly among children [4]. Such oral health education plans can be about oral hygiene or the detrimental effects of poor dietary choices, tobacco use, and excessive alcohol consumption on oral health. Various sources, such as dental professionals, schools, parents, and media, play pivotal roles in disseminating oral health information [5]. By empowering individuals with this knowledge, oral health education not only helps prevent dental problems but also encourages early intervention and treatment, ultimately contributing to improved overall oral health and well-being.

Previous research has focused on the effect of a specific source of information on oral health. However, the question of which one of these sources plays a more important role has remained unanswered. Therefore, the aim of this research article is to determine the relationship between the different sources of oral health information (parents, the internet, television programs, newspapers or magazines, and social media) and the number of decayed, missing, filled teeth (DMFT) among 12- and 15-year-old children living in England in 2013.

## Methods

All codes and metadata related to the study were shared via its OSF repository (https://osf.io/2vazm/) at the time of submission of the manuscript.

### Source of the data

This was a retrospective cross-sectional study using the Child Dental Health Survey (CDHS) 2013 dataset. The CDHS 2013 represents the fifth installment within the sequence of decennial cross-sectional national surveys conducted in the United Kingdom, focusing on children's health. The study encompassed students aged 5, 8, 12, and 15 who were enrolled in mainstream state and independent schools across England, Wales, and Northern Ireland. The survey encompassed a total of 13,628 students. These students were selected in four different sampling stages: 1) sampling regions and local authorities (probability proportional to size), 2) sampling school groups (random sampling without replacement), 3) sampling schools within school groups (simple random sampling), and 4) sampling children within schools (sequential random sampling). Data acquisition for the CDHS 2013 involved multiple methods, including a clinical dental examination, a self-completion questionnaire administered to older children, and a parental questionnaire. In order to achieve approximately 10,000 dental examinations

(2,500 in each age group), the research design called for a sample size of 20,922 pupils. Additional comprehensive information regarding the survey's design has been previously published elsewhere [6].

Possible sources of information in this dataset include parents, television, newspapers and magazines, the internet, and social media. Permanent teeth caries experience (DMFT) was considered the dependent variable; therefore, I only included 12- and 15-year-olds. The other reason was that the exposure variables were only available for 12- and 15-year-olds since the self-completion questionnaire was only given to these age groups. Age, sex, and deprivation level (based on country-specific Indices of Multiple Deprivation or IMD) were used for model adjustment.

## Data analysis

As DMFT is count data, regression models with count data distributions were chosen. Among them, Poisson regression is the most widely used. However, since the data showed over-dispersion, a negative binomial regression was used. Overdispersion was confirmed by comparing the mean (0.19) and standard deviation (0.79) of DMFT. Also, the result of the overdispersion test showed an overdispersion estimate of 1.95 ($p < 0.001$).

# Results

## Descriptive statistics

This dataset included 2,372 children from England. Of these, 51.3% were male, and 48.7% were female. The DMFT was a highly skewed variable with many zeros; therefore, both the median and interquartile range were 0. In Table 1, frequencies and weight-adjusted proportions are reported for age groups, IMD, and sources of information.

## Hypothesis testing

Eight different models were constructed: five for the unadjusted relationship between DMFT and each source of oral health information; one with all sources; one with all sources and adjusted for sociodemographic variables; and the final model. Table 2 depicts the summary for each model. I should note that the numbers for categorical variables are (prevalence) rate ratios. All numbers in Table 2 are back-transformed from their logarithm provided by the statistical software. Here, I scrutinize models 6–8.

The value of the intercept in the negative binomial model stands for the expected (average) value of the count dependent variable when the values of all predictor variables in the dataset are set to 0. Here, the expected value of DMFT when all other variables are 0 is expected to be in the range of 0.17–0.59.

All the following numbers are from adjusted models. Therefore, they are the estimates for the degree of association between each variable and DMFT, holding all other variables in the model constant.

When all sources of oral health information and sociodemographic variables were added in model 7, two sources (internet and social media) and two demographic variables (being of age 15 and female) had a direct relationship with DMFT. It means that schoolchildren using these sources, girls, and those aged 15, had more tooth decay experience (higher prevalence rate ratio). However, three sources, newspapers and magazines, the internet and social media, and sex, did not have a significant relationship with DMFT. Therefore, they were excluded from the final model.

**Table 1. Descriptive statistics of included variables.**

| Characteristic | Number (%) |
|---|---|
| *Sex* | |
| Female | 1222 (48.7) |
| Male | 1150 (51.3) |
| *Age* | |
| 12 years | 1215 (48.6) |
| 15 years | 1157 (51.4) |
| *Index of Multiple Deprivation (England)* | |
| Most deprived quintile | 986 (32.5) |
| 2$^{nd}$ quintile | 448 (20.1) |
| 3$^{rd}$ quintile | 305 (14.4) |
| 4$^{th}$ quintile | 340 (18.2) |
| Least deprived quintile | 293 (14.9) |
| *Source of oral health information* | |
| Parents | 2140 (89.5) |
| Internet | 1042 (43.4) |
| Television programs | 852 (35.5) |
| Newspapers or magazines | 580 (26.2) |
| Social media | 351 (14.3) |
| *DMFT* | |
| 0 | 2071 (90.2) |
| 1 | 183 (6.1) |
| More than 1 | 118 (3.7) |

The final model includes parents and television as sources of oral health information, as well as age and deprivation as sociodemographic variables. This model shows that the expected DMFT for schoolchildren receiving information from their parents was 0.45 times the expected DMFT for schoolchildren who did not receive such information from their parents. Also, schoolchildren who used a television as a source had a 38% lower rate of DMFT on average than those who did not use a television.

Fifteen-year-olds had a 47% higher prevalence rate of DMFT compared with 12-year-olds. Looking at IMD quintiles (categorical variable) reveals a gradient that schoolchildren from less deprived backgrounds had a lower rate of DMFT on average compared to those from the most deprived category. For instance, the two least deprived categories (IMD 5$^{th}$ and 4$^{th}$) had 83% lower prevalence rates of DMFT compared with the most deprived category (IMD 1$^{st}$, the reference group).

The final model formula is as follows:

$$\mathrm{DMFT} = 0.59 + 0.45 X_{parents} + 0.62 X_{TV} + 1.47 X_{age} + 0.17 X_{JMD5} + 0.17 X_{JMD4} + 0.59 X_{JMD3} + 0.63 X_{JMD2} + \varepsilon$$

## Testing model fitness

When comparing the final model with model 7 (including all variables) using the likelihood ratio test, the associated chi-squared value was -6.56 (df = 4, p = 1). Therefore, there is no evidence that the final model underperforms compared with the model with all the variables.

**Table 2. The results of different regression models.**

| Model | Intercept | Parents | Television | Newspapers | Internet | Soc Media | Age 15 | Female | IMD 5th | IMD 4th | IMD 3rd | IMD 2nd |
|---|---|---|---|---|---|---|---|---|---|---|---|---|
| 1 | 0.35 (0.23–0.56) *** | 0.47 (0.29–0.75) ** | — | — | — | — | — | — | — | — | — | — |
| 2 | 0.20 (0.17–0.25) *** | — | 0.76 (0.54, 1.07) | — | — | — | — | — | — | — | — | — |
| 3 | 0.19 (0.16–0.23) *** | — | — | 0.87 (0.61, 1.27) | — | — | — | — | — | — | — | — |
| 4 | 0.17 (0.14–0.22) *** | — | — | — | 1.17 (0.84, 1.61) | — | — | — | — | — | — | — |
| 5 | 0.17 (0.14–0.20) *** | — | — | — | — | 1.67 (1.10, 2.61) * | — | — | — | — | — | — |
| 6 | 0.34 (0.22–0.55) *** | 0.46 (0.28–0.73) ** | 0.74 (0.52–1.05) * | 0.90 (0.62–1.32) | 1.18 (0.82–1.69) | 1.68 (1.07–2.68) * | — | — | — | — | — | — |
| 7 | 0.51 (0.30–0.92) * | 0.44 (0.26–0.70) *** | 0.59 (0.41–0.85) ** | 0.87 (0.60–1.28) | 1.20 (0.83–1.73) | 1.52 (0.97–2.41) | 1.43 (1.04–1.98) * | 1.13 (0.82–1.56) | 0.19 (0.10–0.33) *** | 0.17 (0.10–0.30) *** | 0.64 (0.40–1.03) | 0.61 (0.40–0.93) * |
| 8 (final) | 0.59 (0.36–1.01) | 0.45 (0.27–0.72) *** | 0.62 (0.44–0.88) ** | — | — | — | 1.47 (1.06–2.03) * | — | 0.17 (0.09–0.30) *** | 0.17 (0.10–0.29) *** | 0.59 (0.37–0.95) * | 0.63 (0.42–0.96) * |

*: P<0.05

**: P<0.01

***: P<0.001

## Checking model assumptions

Negative binomial models assume the conditional means are not equal to the conditional variances. This inequality is captured by estimating a dispersion parameter that is held constant in a Poisson model. Thus, the Poisson model is actually nested in the negative binomial model [2]. I used a likelihood ratio test to compare these two and test this model assumption. The associated chi-squared value was 626.56 with 4 degrees of freedom (p<0.001). This strongly suggests the negative binomial model, estimating the dispersion parameter, is more appropriate than the Poisson model.

## Discussion

This study showed a decrease in the prevalence rate of DMFT in students whose sources of oral health information were their parents (0.45) or television (0.62). Fifteen-year-olds and students from deprived areas had a higher prevalence rate. The prevalence rate for newspaper was lower than 1, and for the Internet and social media, were higher than 1; however, these were not statistically significant.

The lower prevalence rate for parents can be linked to socioeconomic status, as parents who are more vigilant about their children's oral health and provide guidance tend to experience less deprivation [7, 8]. Higher socioeconomic status affords parents improved access to education and oral health information, making them more aware of the significance of proper oral hygiene and dental care practices [9]. Consequently, these parents often impart their knowledge to their children, emphasizing the importance of regular dental check-ups, correct brushing techniques, and a healthful diet.

Moreover, socioeconomic status can significantly shape dietary choices within households. Families with higher incomes tend to enjoy greater access to nutritious foods and snacks,

thereby contributing to enhanced oral health outcomes [10]. Conversely, lower-income families may find themselves relying more on processed foods and sugary snacks, recognized culprits in dental decay.

In this study, an intriguing pattern emerged regarding the sources of oral health information among children. Those who relied on television as their primary source were less likely to experience dental caries, while the Internet and social media were associated with a higher prevalence rate of dental caries (DMFT). This divergence in outcomes can be attributed to several factors related to the nature and oversight of these information sources.

Television, as a medium, offers better controllability when compared to the Internet and social media. Television programs operate within a centralized framework and typically undergo more rigorous quality checks before content is broadcast. Consequently, the information presented on television tends to be curated, reliable, and accurate, contributing to better dental health outcomes.

Furthermore, the role of parental involvement cannot be understated. Parents often monitor and guide their children's television viewing habits, ensuring that they are exposed to educational content related to oral health [11]. This proactive engagement by parents serves to reinforce the oral health messages conveyed through television, encouraging children to adopt and practice good oral hygiene habits.

In contrast, the Internet and social media represent decentralized systems where anyone can post information, much of which may be inaccurate or misleading [12]. Unfortunately, children's use of these platforms is not always subject to parental supervision or guidance. This lack of oversight can result in children accessing unreliable sources of information and potentially adopting less effective oral health practices. For example, children might be influenced by friends or online influencers who engage in behaviours like excessive sugary food consumption or neglecting oral hygiene.

There are few studies conducted on the relationship between source of oral health information and dental caries. In a study conducted in 1989 in the United States, whereas the extent of oral health knowledge was not related to mean DMFT, children with the highest level of oral health information reported getting their information from parents and family [13]. This can be related to the findings of our study that parents were an effective source of oral health information.

## Implications

Our study showed the importance of oral health information received from parents and television for schoolchildren. This information may motivate children to practice oral hygiene more seriously compared to those whose source of oral health information is other media such as the internet or social media. Therefore, oral health professionals, researchers, and policymakers may scrutinize the potential of these two sources for reducing the burden of dental caries.

## Limitations

Because of the excessive number of zeros in the DMFT variable in this dataset, it is suggested to use a zero-inflated negative binomial regression [3]. Zero DMFT may be because of different processes, such as receiving fluoridated water. A standard negative binomial model does not distinguish between processes that lead to zeros, but a zero-inflated model allows for and accommodates this complication. The same as above was done here for comparing the final model with a zero-inflated one. The associated chi-squared value was 2.51 (df = 4, p = 0.643). Therefore, using a non-zero-inflated model is preferable due to its simplicity.

## Conclusion

Children whose primary source of oral health information was their parents or television had a lower DMFT. On the contrary, using the Internet or social media as the source of oral health information was associated with higher caries experience among schoolchildren.

## Author Contributions

**Conceptualization:** Ahmad Sofi-Mahmudi.

**Data curation:** Ahmad Sofi-Mahmudi.

**Formal analysis:** Ahmad Sofi-Mahmudi.

**Investigation:** Ahmad Sofi-Mahmudi.

**Methodology:** Ahmad Sofi-Mahmudi.

**Project administration:** Ahmad Sofi-Mahmudi.

**Resources:** Ahmad Sofi-Mahmudi.

**Software:** Ahmad Sofi-Mahmudi.

**Supervision:** Ahmad Sofi-Mahmudi.

**Validation:** Ahmad Sofi-Mahmudi.

**Visualization:** Ahmad Sofi-Mahmudi.

**Writing – original draft:** Ahmad Sofi-Mahmudi.

**Writing – review & editing:** Ahmad Sofi-Mahmudi.

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
