## [Decision Letter · Decision Letter 0]

12 Dec 2023

PONE-D-23-31617The relationship between the source of oral health information and dental caries: Findings from Child Dental Health Survey 2013 in EnglandPLOS ONE

Dear Dr. Sofi-Mahmudi,

Thank you for submitting your manuscript to PLOS ONE. After careful consideration, we feel that it has merit but does not fully meet PLOS ONE’s publication criteria as it currently stands. Therefore, we invite you to submit a revised version of the manuscript that addresses the points raised during the review process.

We look forward to receiving your revised manuscript.

Kind regards,

Yolanda Malele-Kolisa, BDS, MPH, MDent, PhD

Academic Editor

PLOS ONE

Journal Requirements:

Reviewers' comments:

Reviewer's Responses to Questions

**Comments to the Author**

1. Is the manuscript technically sound, and do the data support the conclusions?

Reviewer #1: Yes

Reviewer #2: Partly

2. Has the statistical analysis been performed appropriately and rigorously? 

Reviewer #1: Yes

Reviewer #2: Yes

3. Have the authors made all data underlying the findings in their manuscript fully available?

Reviewer #1: Yes

Reviewer #2: Yes

4. Is the manuscript presented in an intelligible fashion and written in standard English?

Reviewer #1: Yes

Reviewer #2: No

5. Review Comments to the Author

Reviewer #1: The author needs to describe the findings in the results as much as the description of the models in the results section, ensure that table 1 is in one page. The research ethics were followed promptly as this is a secondary data analysis from a primary survey data source. The author has to elaborate on that in a scientific way.

Reviewer #2: Title and Abstract:

• Gives a brief about what the article is about, significance of the study while the aim is stated clearly.

Introduction :

• Scientific background and rational stated

• Objectives not clearly stated

Methods:

• Study design not stated.

• Setting or data source described.

• Rationale for selecting the age groups (participants) is not explained.

• Method of sampling not clearly described.

• Sample size not mentioned, it is only mentioned in the results.

• No explain of how the study size was arrived at.

• Statistical methods are well described.

Discussion:

• No clear discussion of the implications of the results to the scientific field

• The authors did not clearly provide reasoned arguments for their interpretation of their results.

• No comparison with previous studies ,no highlight of similarities or differences.

General comments:

• Grammatical errors, need editing of the article.

• Write using third person.

• Data Zip file was empty

6. PLOS authors have the option to publish the peer review history of their article (what does this mean?). If published, this will include your full peer review and any attached files.

Reviewer #1: No

Reviewer #2: **Yes: **Cathrine Batseba Malau

---

## [Author Response · Author response to Decision Letter 0]

12 Mar 2024

Reviewer #1: The author needs to describe the findings in the results as much as the description of the models in the results section, ensure that table 1 is in one page. The research ethics were followed promptly as this is a secondary data analysis from a primary survey data source. The author has to elaborate on that in a scientific way.

Authors: Thanks for your comment. We amended the manuscript accordingly.

Reviewer #2: 

Title and Abstract:

• Gives a brief about what the article is about, significance of the study while the aim is stated clearly.

Authors: Thanks for your comment.

Introduction :

• Scientific background and rational stated

• Objectives not clearly stated

Authors: Thanks for your comment. We revised the objectives.

Methods:

• Study design not stated.

Authors: Thanks for your comment. We added the study design.

• Setting or data source described.

• Rationale for selecting the age groups (participants) is not explained.

Authors: Thanks for your comment. We added the rationale for selecting these age groups. These age groups were the only age groups with the data.

• Method of sampling not clearly described.

Authors: Thanks for your comment. We added the sampling method.

• Sample size not mentioned, it is only mentioned in the results.

Authors: Thanks for your comment. We already mentioned sample size under the “Sources of the data” in the Methods section.

• No explain of how the study size was arrived at.

Authors: Thanks for your comment. We added the details about the sample size.

• Statistical methods are well described.

Discussion:

• No clear discussion of the implications of the results to the scientific field

Authors: Thanks for your comment. We added more implications to the Discussion.

• The authors did not clearly provide reasoned arguments for their interpretation of their results.

Authors: Thanks for your comment. We already had provided the arguments for our interpretations (e.g., the 2nd and 3rd paragraph of the Discussion).

• No comparison with previous studies ,no highlight of similarities or differences. 

Authors: Thanks for your comment. We added a paragraph in this regard.

General comments:

• Grammatical errors, need editing of the article. 

• Write using third person.

• Data Zip file was empty

Authors: Thanks for your comment. We checked all the errors and amended the manuscript accordingly.

---

## [Decision Letter · Decision Letter 1]

5 Apr 2024

The relationship between the source of oral health information and dental caries: Findings from Child Dental Health Survey 2013 in England

PONE-D-23-31617R1

Dear Dr. Sofi-Mahmudi,

We’re pleased to inform you that your manuscript has been judged scientifically suitable for publication and will be formally accepted for publication once it meets all outstanding technical requirements.

Kind regards,

Yolanda Malele-Kolisa, BDS, MPH, MDent, PhD

Academic Editor

PLOS ONE

Additional Editor Comments (optional):

ACCEPT SUBJECT TO THE REVISIONS BELOW:

5,8,12 and 15 : This information is not true, you only added 12 and 15 year olds as later mentioned in the paragraph below.

This was the calculated sample size? 20922 This means out of the intended 20922 you managed to review 2372 patient records?

Please discuss this and the implications of the findings considering the ratio rate equivalent to response rate. Add to the limitations.

Reviewers' comments:

Reviewer's Responses to Questions

**Comments to the Author**

1. If the authors have adequately addressed your comments raised in a previous round of review and you feel that this manuscript is now acceptable for publication, you may indicate that here to bypass the “Comments to the Author” section, enter your conflict of interest statement in the “Confidential to Editor” section, and submit your "Accept" recommendation.

Reviewer #2: All comments have been addressed

2. Is the manuscript technically sound, and do the data support the conclusions?

Reviewer #2: Yes

3. Has the statistical analysis been performed appropriately and rigorously? 

Reviewer #2: Yes

4. Have the authors made all data underlying the findings in their manuscript fully available?

Reviewer #2: Yes

5. Is the manuscript presented in an intelligible fashion and written in standard English?

Reviewer #2: Yes

6. Review Comments to the Author

Reviewer #2: The authors have adequately addressed the concerns raised in the previous version. It is recommended that the manuscript go through language editing as there are still come areas where words like "WE" are still used in the following sections, please rephrase:

Source of data

Hypothesis testing

Checking model assumptions.

7. PLOS authors have the option to publish the peer review history of their article (what does this mean?). If published, this will include your full peer review and any attached files.

Reviewer #2: **Yes: **Dr Cathrine Malau

---

## [Editor Report · Acceptance letter]

22 Jun 2024

PONE-D-23-31617R1 

PLOS ONE

Dear Dr. Sofi-Mahmudi, 

I'm pleased to inform you that your manuscript has been deemed suitable for publication in PLOS ONE. Congratulations! Your manuscript is now being handed over to our production team.

Kind regards, 

on behalf of

Prof Yolanda Malele-Kolisa 

Academic Editor

PLOS ONE